# Thermo-Mechanical Coupling Analyses for Al Alloy Brake Discs with Al_2_O_3_-SiC_(3D)_/Al Alloy Composite Wear-Resisting Surface Layer for High-Speed Trains

**DOI:** 10.3390/ma12193155

**Published:** 2019-09-27

**Authors:** Lan Jiang, Yanli Jiang, Liang Yu, Hongliang Yang, Zishen Li, Youdong Ding

**Affiliations:** 1Key Laboratory for Ecological Metallurgy of Multimetallic Mineral (Ministry of Education), Northeastern University, Shenyang 110819, China; jiangl@smm.neu.edu.cn (L.J.); yanghlneu@163.com (H.Y.); lizsneu@163.com (Z.L.); dingydneu@163.com (Y.D.); 2School of Metallurgy, Northeastern University, Shenyang 110819, China; 3Key Laboratory of New Processing Technology for Nonferrous Metals & Materials, Ministry of Education, Guilin University of Technology, Guilin 541004, China

**Keywords:** Al_2_O_3_-SiC_(3D)_/Al alloy composite, wear-resisting layer, brake disc, finite element method, thermal stress

## Abstract

In the present work, a theoretical model of three-dimensional (3D) transient temperature field for Al alloy brake discs with Al_2_O_3_-SiC_(3D)_/Al alloy wear-resisting surface layer was established. 3D transient thermo-stress coupling finite element (FE) and computational fluid dynamic (CFD) models of the brake discs was presented. The variation regularities of transient temperature and internal temperature gradient of the brake discs under different emergency braking conditions were obtained. The effects of initial braking velocity (IBV) and thickness of Al_2_O_3_-SiC_(3D)_/Al alloy composite wear-resisting layer on the maximum friction temperature evolution of the disc were discussed. The results indicated the lower temperature and thermal stress distributed uniformly on the wear-resisting surface, which was dominated by high conductivity and cooling ability of the Al alloy brake disc. The maximum friction temperature was not obviously affected by the thickness of the wear-resisting layer. The maximum friction temperature of the brake discs increased with the increase of the IBV, the maximum friction temperature and thermal stress of the brake discs is about 517 °C and 192 MPa at IBV = 97 m/s considering air cooling, respectively. The lower thermal stress and fewer thermal cracks are produced during the braking process, which relatively decrease the damage. The friction behavior of the tribo-couple predicted using FE method correlated well with the experimental results obtained by sub-scale testing.

## 1. Introduction

The brake system is a key component for high-speed trains. To guarantee a safe, steady, and durable brake system, brake disc and pad materials generally should maintain a stable and reliable coefficient of friction (COF), irrespective of braking conditions, ambient temperatures, and humidity [1,2,3]. Brake systems need to meet the requirement of stable braking performance and light weight. Unsprung weight can be reduced effectively by using an Al metal matrix composite (Al-MMC) brake disc [4]. The particulate reinforced metal matrix composite SiC_P_/Al and Al_2_O_3(P)_/Al (MMC) brake discs have been investigated for lightweight vehicular applications but have not widely been utilized due to some issues with uniformity of distribution of SiC or Al_2_O_3_ particulates, residual porosity, formation of undesirable phases such as Al_3_C_4_, and high temperatures incurred during braking that degrade the disc’s integrity and performance [5]. In contrast, 3-3 interpenetrating composites (IPC), which are also called co-continuous composites, consisting of three-dimensionally continuous matrices of discrete metal and ceramic phases may provide novel advantages for wear resistance applications as they could offer a higher load bearing capacity than the conventional Al-MMCs [6]. Some researchers [7,8,9] have studied the wear behavior of Al-MMC IPCs and compared it with conventional Al-MMCs, indicating that the former is significantly better [10]. The SiC_3D_/Al alloy and Al_2_O_3(3D)_/Al alloy (3D means three-dimensional network structure) IPCs have been applied widely to dry friction and wear applications, since they show excellent friction and wear performance, low density, and high thermal conductivity [11]. Many simulation results demonstrated that the use of Al-MMCs for the wear-resisting layer and ductile Al alloy for the backing plate can lead to the development of lightweight brake discs to replace the traditional brake discs made of cast iron and steel [12]. 

In our previous work, we prepared a new class ventilated Al alloy brake disc of sandwich type structure with wear-resisting SiC_3D_/Al friction surface layers [13]. The presence of hard SiC reticulated porosity ceramics (RPC) serves to delay the transition from mild to severe wear, and improve high-temperature capabilities under different emergency braking conditions [14,15]. However, the high hardness of SiC RPC results in high wear rate of the friction pads. 

We reduced the hardness of SiC RPC by adding Al_2_O_3_ phase and reduced the volume ratio of RPC in the composites in order to reduce the wear losses of the friction pads, which make tribo-couples show better matching friction and wear performance. We designed the Al alloy brake discs with variation in thickness of Al_2_O_3_-SiC_(3D)_/Al alloy composite wear-resisting surface layers.

Brake discs are exposed to large thermal stress during emergency braking conditions due to the temperature gradient, which may lead to thermal crack, brake fade, and bearing failure [16]. In order to verify the reliability of this new brake disc and secure braking performance stability, in this paper, a coupling approach between 3D finite element (FE) solid heat conduction and computational fluid dynamic (CFD) fluid heat transfer was employed to investigate temperature and thermal stress distributions of the brake disc [17,18,19]. Experimental data was used to validate the simulation result. The reason for the thermal cracks on the disc surface was discussed as well.

## 2. Geometry Design of the Brake Disc

The 3D calculation model was developed by Solidworks2018 software for analysis as shown in Figure 1. 

The weight of the Al alloy brake disc is about 22 kg compared to 100 kg of iron brake disc. The friction layer was divided into 12 blocks. The wear-resisting surface layers are made of Al_2_O_3_-SiC_(3D)_/Al alloy composite and other parts of the brake disc are made of Al alloy. The disc body was designed with 24 cooling ribs structures to increase convective surface. To find out the relationship between thickness of the wear-resisting surface layer and maximum friction temperature of the brake disc, the thickness of wear-resisting layer was set as 2, 4, and 6 mm, respectively. The material parameters of the brake disc are listed in Table 1. The basic data of the brake disc are listed in Table 2.

The microstructure of Al2O3-SiC(3D)/Al composites is complex, is difficult to model according to the real structure of materials, and it is difficult to mesh it by finite element method (FEM) as well. Therefore, the mixing law of physical quantities of inherent properties is adopted in this study. That is to say, if the property of composite system is Γ, the properties and volume fractions of any phase in the composite are Γ_i_ and V_i_ respectively, then the material of composite material is $\;\Gamma ={\Sigma \Gamma }_{i}V_{i}$. The properties of Al_2_O_3_-SiC_(3D)_/Al composites listed in Table 1 are calculated by mixing rate based on properties and volume percentages of Al_2_O_3_-SiC_(3D)_ silicon carbide and Al alloy, respectively.

The specific heat capacity C_p_ and heat conductivity H_c_, changing with temperature, will have a serious influence on the thermal simulation. Therefore, temperature determined specific heat capacity C_p_ and heat conductivity H_c_ of the Al_2_O_3_-SiC_(3D)_/Al composites and Al alloy matrix were measured, respectively. 

According to the standard method (ASTM E 1461), the heat conductivity H_c_ of the Al_2_O_3_-SiC_(3D)_/Al composites and Al alloy matrix were measured by laser flash method. The laser thermal conductivity tester was FL4010. The samples were processed by water cutting. The sample size was round, the diameter was 12.7 mm, and the thickness was 3 mm. According to the standard method (ASTM E 1269, Standard Test Method for Determining Specific Heat Capacity by Differential Scanning Calorimetry), specific heat capacity C_p_ of the Al_2_O_3_-SiC_(3D)_/Al composites and Al alloy matrix with temperature change were measured by DSC method. The specific heat capacity tester was DSC Q2000. The weight of the samples was 30 mg. 

As shown in Figure 2, the specific heat capacity C_p_ of Al_2_O_3_-SiC_(3D)_/Al composites and Al alloy matrix increase from 900 to 1350 J·kg^−1^·K^−1^, and from 980 to 1390 J·kg^−1^·K^−1^ with temperatures increasing from 140 to 460 °C (red line and the red y-coordinate), respectively. The heat conductivity H_c_ of Al_2_O_3_-SiC_(3D)_/Al composites and Al alloy matrix increase from 152 to 162 w·m^−1^·K^−1^, and from 160 to 168 w·m^−1^·K^−1^ with temperatures increasing from 140 to 460 °C (blue line and the blue y-coordinate), respectively.

## 3. Simulation Modeling

### 3.1. Heat Flux Input

During the braking process, the kinetic energy of a moving vehicle is converted to thermal energy through friction heating between the brake disc and the pads. Frictional heat is generated on the surface of the brake disc and brake pads. The heat flux q entering the disc is calculated by Equations (1) and (2) [12]:(1)$$Q(t)=W=\frac{1}{2}M\left(V_{0}^{2}-V_{t}^{2}\right),$$
(2)$$q\left(t\right)=\frac{dQ}{Adt}=\frac{d\left(\frac{1}{2}M\left(V_{0}^{2}-V_{t}^{2}\right)\right)}{Adt}=-\eta Ma\left(V_{0}-at\right)/A,$$
where M  is the shaft mass of the single disc. *V_0_*, and *V_t_* represent the initial braking velocity (IBV) and velocity at time t of the vehicle, respectively. a is acceleration,  η is relative braking energy absorbed by the brake disc. The value of η is in terms of the material properties of the brake disc and pad, the thermal conductivity of material of brake pad is generally smaller than that of the disc, in this paper, η was defined as 0.9. *A* is contact friction area.

### 3.2. Heat Transfer and Thermal Stress

The equation for calculating the heat transfer and thermal stress of brake disc are described in greater depth in [18].

### 3.3. Thermal Flow 

The Solidwork2018 Flow package automatically calculates heat transfer coefficient at the wall boundary using Equation (3) [19].
(3)$$h_{c}=\frac{q_{w}}{T_{b}-T_{nw}},$$
where $h_{c}$ is heat transfer coefficient, $q_{w}$ is heat flux at the wall boundary, *T*_b_ is the specified boundary temperature (that is, outside the fluid domain), and $T_{nw}$ is the temperature at the internal near wall boundary element center node. 

### 3.4. Boundary Condition

#### 3.4.1. Air Convection

With the help of Solidwork2018 Flow Simulation codes, the air around the brake disc was characterized, and heat convection coefficients were calculated and used as the boundary condition in thermal analysis [20]. The CFD model, brake disc, hub, and fluid domain, was established for calculation basing on the real working conditions of the brake disc. A cuboid fluid domain was established with size of 1500 × 1500 × 1500 mm shown in Figure 3, which was large enough to eliminate the effects of boundaries on flow and heat transfer nearby the brake disc. The air flows from the left of the model, and the air flow rate will decrease with the decrease of the train speed.

Local rotation domain was induced for simulation of the rotated air flow around the brake disc. A range of angular speeds was applied to the brake disc and hub. The velocity field and the pressure profile were assigned on the boundary surface, and open boundaries with 0 relative pressure were used for the upper, lower, and radial ends of the domain. Air around the disc was 25 °C and reference pressure was 1 atm. Surface roughness of the brake disc was 100 μm. The 3D thermo-fluid coupling model (including brake disc, shaft, rotating air domain) was constructed for analysis. The disc model was attached to an adiabatic shaft. The mesh of 542,286 cells for the brake system and 341,913 cells for the cooling air zone were chosen. The κ−ε model was selected. We conducted the transient temperature and thermal stress analyses of the brake disc using the Solidworks 2018 simulation software based on FE method. The initial temperature was 60 °C. Total time of simulation was 200 s, the time step increment was t = 1 s, and increment of minimal initial time was 0.5 s. The temperature field simulation was carried out firstly, and then the stress field investigation was performed based on the thermal boundary conditions [21]. The multi physical field coupling diagram of the disc brake is shown in Figure 4.

The researchers simplified the complex *h_c_* of friction pairs in the braking process to a constant value. *h_c_* does not vary with temperature and velocity. *h_c_* is closely related to the IBV and the shape of heat dissipation design of the friction pair. That is to say, *h_c_* at different positions of the friction pair is actually different, *h_c_* is a non-linear data varying with the braking process. The sequential coupling method was adopted in our study to realize the coupled calculation method of frictional heat-stress-flow field. Figure 5 shows the FEM-CFD coupling calculation flow chart of the disc brake. Firstly, the thermal-mechanical coupling analysis of the friction pair model is carried out. Then the data of *h_c_* calculated by CFD software were loaded on the model of thermo-mechanical coupling analysis for the second coupling calculation. As *h_c_* calculated by CFD is a function of braking time, the *h_c_* of different positions of friction pairs is different. Therefore, the temperature field, stress field, and flow field can be obtained by coupling the *h_c_* function with the thermo-mechanical calculation data again, which can accurately predict the thermal stress concentration area in the brake disc.

#### 3.4.2. Thermal-Stress Calculation

The convection analysis data obtained from CFD simulation was directly imported into Solidwork2018 simulation software in the form of ‘fld’ format file as heat dissipation boundary conditions in thermal analysis [22]. As the stress was analyzed, the brake system components needed to be proper contact constraints. The disc was put together rigid where the disc was screwed onto the hub. The initial temperature was 60 °C. Total simulation time was 300 s, time step increment was 0.5 s, and increment of minimal initial time was 0.25 s. The overall meshing was tuned to be appropriate and more refined in the contact zone between the brake disc and pad. Finally, 223,551 nodes, and 132,702 elements were chosen for the calculation as shown in Figure 6. 

## 4. Experimental Setup

### 4.1. Al_2_O_3_-SiC_(3D)_ Preparation 

Commercial SiC powders (≥99%, D_50_ = 0.6 µm) and Al_2_O_3_ powders (99.8%, D_50_ = 2 µm) were used as the two starting materials for preparing Al_2_O_3_-SiC_(3D)_ RPCs. First, polyvinyl alcohol PVA (Guilin Baer chemical reagents Co., Ltd., China, binder), sodium carboxymethyl-cellulose (Guilin Baer Chemical Reagents Co., Ltd., China CMC, thickening agent), and Dolapix CE-64 (Zschimmer and Schwarz, dispersant) were used as the additives. These additives were added to deionized water. The weight percentages of Dolapix CE-64, CMC and PVA based on the ceramic powder were 2, 0.5, and 1.5 wt% in the Al_2_O_3_-SiC slurry, respectively. After stirring to obtain premixed solutions, Al_2_O_3_ and SiC powders with a weight ratio of 85:15 were added to prepare the slurry with a solid content of 55 vol.%. The slurry was ball-milled for 4 h. The homemade 3D extrusion printing machine was used to form the green bodies with the as-prepared slurry. Figure 7 presents a schematic diagram 3D extrusion printing machine for fabricating Al_2_O_3_-SiC_(3D)_ green bodies. Next, the prepared green bodies were dried at 25 °C for 24 h and further at 120 °C for 24h. The dried samples were subjected to pressureless sintering in air at 1450 °C for 1h to obtain Al_2_O_3_-SiC_(3D)_ RPCs.

### 4.2. Al_2_O_3_-SiC_(3D)_/Al IPC Preparation

Al_2_O_3_-SiC_(3D)_/Al composites were prepared by infiltration of a molten Al alloy liquid (chemical compositions, 2.4 wt% Cu, 9.2 wt% Si, 0.3 wt% Fe, 0.3 wt% Mn, 1.8 wt% Mg, 0.3 wt% Zn, 0.5 wt% Sc, 0.3 wt% Ti, balance Al) into the Al_2_O_3_-SiC_(3D)_ RPCs using a low-pressure casting method. The T6 heat treatment for the brake discs includes solution treatment quench and artificial aging. In solution phase the brake discs were heated to 525 °C for 8 h, then were quenched less than 15 s in 60 °C water. Finally, in the artificial aging, the brake discs were heated to 175 °C for 8 h. The processing and physical properties of the composites were described in greater depth in [12,13]. The microstructures of the composites were characterized using optical metallography (OM, GX71, Olympus, Japan), and scanning electron microscopy (SEM, SU4800, Shimadzu, Tokyo, Japan).

### 4.3. Frictional Experiment

Full-sized bench tests for brake friction pairs of the high-speed railway are expensive. The cost of experiment can be reduced, and reliable experimental data can be obtained quickly, by using the sub-scale testing machine. In this study, to verify the simulation, experiment investigations were carried out on the MM3000 sub-scale testing machine shown in Figure 8 (Xi’an ShunTong Technical Research Institute, Xi’an, China) at a room temperature and a relative humidity of 45%. A MM3000-type machine can operate a rotating ring-on-pin braking test. After setting the simulated moment of inertia, the highest rotational velocity of the rotating chuck, and the positive pressure of the static chuck during the brake operation, a braking test was performed automatically with the MM3000. The axial force was detected with a pressure sensor and the frictional torque was measured by the pulling force sensor. An infrared thermometer recorded the surface temperature of friction pairs.

Figure 9 shows the schematic diagram of the rotating brake ring and powder metallurgy pins used for the sub-scaled test. The size of the brake ring with wear-resisting layer thickness of 4 mm made out of Al_2_O_3_-SiC_(3D)_/Al alloy composite (black arrow indicates) has an outer diameter of 180 mm, and inner diameter of 158 mm. The effective radius of the braking ring is 84.5 mm. The counterparts (pins) were machined from powder metallurgy (PM) brake pads provided by Knorr Corporation with thickness of 8 mm. Three PM pins were fixed on the stationary ring (red arrow indicates). For obtaining an accurate value of the temperature, a thermocouple and infrared radiation thermometer were used simultaneously. The distance between the point for measuring temperature of thermocouple and the friction surface was 0.5 mm. 

The applied initial rotation speed was 55~97 m/s in brake disc effective radius linear velocity, corresponding to the IBV of 200~350 km/h of the high-speed trains. 

In order to accurately compare experimental and simulated data, the energetic approach for the simulation, and the energetic approach for the experimental setup, should be imposed the same heat flux at the contact surface. Therefore, we used the reduced scale rule Equation (4) [23], the test samples, and test schemes were prepared as well. 

The reduced scale rule has two principles. The first one is that the energy absorbed by the unit area of the friction surface of the experimental ring and the brake disc is the same. The second one is to increase the angular velocity of the experimental ring so that the linear velocity of the rotation of the experimental ring is the same as that of the rotation of the brake disc.
(4)$$\frac{0.5\;\times \;MV_{0}{}^{2}\times \;\eta }{A_{1}}=\frac{0.5\;\times \;I\;\times \;\omega^{2}}{A_{2}},$$
where M is shaft mass of single disc, M = 4.4 × 10^3^ kg; V_0_ is IBV, (m·s^−1^); I is initial-mass disc of sub-scale testing machine, I = 0.80 kg·m^2^; ω is angular velocity of sub-scale testing machine, rad/s; A_1_ is total friction area on both sides of train brake disc, A_1_ = 4.48 × 10^5^ mm^2^; A_2_ is friction area of brake ring used for sub-scale testing, A_2_ = 5.83 × 10^3^ mm^2^;  η is in terms of the material properties of the brake disc and pad, η  = 0.90. After calculation, the experimental parameters for sub-scale testing are listed in Table 3.

During the tests, the initial-mass disc of 0.80 kg·m^2^ was rotated up to 6267–10958 rpm before braking, then the 0.80 MPa was applied on the pin through a gas operated piston until the flywheel completely stopped. 

The microstructure of the composites was characterized using optical metallography. The microstructure of the composites was characterized using optical metallography (OM, GX71, Olympus, Japan). The worn surface was observed by confocal laser scanning microscope (CLSM, Axio-Imager LSM-800, Zeiss, Oberkochen, Germany).

## 5. Results and Discussion

### 5.1. CFD Analysis

The air flows through the brake disc were determined by CFD simulation [24]. It is obvious that high-speed rotation of the brake disc with symmetrical ribs can drive the rotation of the surrounding air flow (Figure 10). An air turbulence vortex was observed. The ribs speed up the air flow, which effectively improve convection heat transfer and reduce the temperature of the friction surface. 3D streamlines of the air around the brake disc were calculated at 5 s after start of braking. It is clear that the faster the rotation of the brake disc, the greater the heat dissipation capacity can be gained. Convective heat transfer coefficient distribution was obtained by CFD [25].

### 5.2. Temperature Analysis

The variation of the maximum friction temperature of the brake discs with different thickness of wear-resisting layer versus braking times is presented in Figure 11. The value of temperature of the brake disc without considering air cooling is about 18% higher than that those of considering air cooling at IBV 97 m/s. By convection cooling of the ribs, heat dissipates to the air effectively. The temperature rose sharply in the beginning stage of braking process followed by a fall of temperature decrease. Furthermore, the maximum friction temperature was not obviously affected by the thickness of wear-resisting layers. Considering the manufacturing cost and processing technology, the 4 mm thickness of the wear-resisting layer is appropriate for preparing the brake disc. A good agreement between simulation results and experiment results shows that the numerical simulation is reliable.

Figure 12a–d present the temperature distribution in the brake disc at different stages of braking operation (IBV 97 m/s, thickness of wear-resisting layer 4 mm, under the braking pressure 22 kN). In general, the uniform distribution of temperature field of the brake disc was obtained due to the high conductivity of the Al alloy brake disc. Temperature is uniformly distributed on the wear-resisting layer. Temperature distribution at the brake disc was not only determined by the thermal capacity, but the heat conductivity of the brake disc material. At 12 s (Figure 12a), the enormous friction heat generated between the friction couple is transmitted to the brake disc in very short time at the initial stage. At 30 s (Figure 12b), heat energy is transferred to the back of the disc rapidly which reduces the accumulation of the friction thermal energy because of the excellent conductivity of the brake disc material, and the maximum friction temperature reaches to about 400 °C at 30 s. The temperature distributes symmetrically and uniformly on the both sides of the disc (Figure 12b). The maximum friction temperature reaches up to 517 °C at 63 s (Figure 12c). The heat energy is mainly concentrated on the wear-resisting layer surfaces, on which some hot spots were observed. Formation of the hot spots seen in the simulation results is attributed to the heterogeneous heat dissipation of the brake disc. However, hot spots disappear in a relatively short time. The maximum friction temperature drops down to 475 °C at 83 s, the moment the brake disc stopped turning (Figure 12d). 

Figure 13 shows the relationship between transient temperature and IBV 55~97 m/s under braking pressure 22 kN. It can be seen that temperatures rapidly increase to the maximum in about 40~60 s, then decrease from 60 to 160 s, and finally, smoothly decrease to 60 °C after 120~240 s. The maximum friction temperature of brake disc increases from 338 to 517 °C with IBV increasing from 55 to 97 m/s. The heat flux and convective heat-transfer reach the dynamic balance as marked by the black dotted line. The balance is dominant among heat flux, convective heat-transfer, and heat conduction [26]. In zone A (lavender region, braking time from 0 to 40 s), heat flux is higher than the convective heat-transfer of the brake disc. On the contrary, in zone B (green region, braking time from 40 to 240 s), heat flux is lower than the convective heat-transfer of the disc.

Figure 14 shows the simulated temperature in the circumference and radial direction versus time at different radii (IBV 97 m/s, thickness of wear-resisting layer 4 mm). Figure 14a shows the temperature evolution at intersecting surfaces of the disc at t 63 s. The maximum temperature reaches up to 509 °C at point a, 389 °C at point b, and 296 °C at point c, respectively. The maximum temperature at the friction surface is higher by about 200 °C than temperature at the back of the disc. It means temperature in the contact region is higher than that in the non-contact region. On the friction surface, along the radius direction from outer to the inner of the disc, the points were marked as 1, 2, 3, 4, respectively, shown in Figure 14b. It is clearly presented that the temperature of the outer/inner edge of the disc increases nearly monotonously with braking time and reaches 438 °C at 52 s (point 1) and 426 °C at 68 s (point 3), respectively. However, at point 2 located in the central equivalent radius, the max temperature is 517 °C at 63 s. Furthermore, the moving heat source has an influence on the fatigue stress and thermal cracking caused by the heat flux, shown in the Figure 14c from A to B on the friction surface. The direction along the friction surface to the back of the disc is labeled as a, b, c, respectively shown in the Figure 14d. Simulated results indicate that the temperature decreases from the point a to the point c (Figure 14a). Simulated results indicate the great temperature gradients are generated obviously between the friction surface and the interior disc body, or between the outer edge and inter edge of the disc, resulting in the compressive stress in the radial and circumferential direction on the surface [27].

### 5.3. Stress Analysis

Stress in the circumference and radial direction versus time at different radii is shown in Figure 15. It is found that stress evolution on the friction surface is concordant with the temperature evolution, namely, both increase after braking application and then decrease to the stable level in the final braking stage [28]. Circumferential stress is obviously greater than the radial stress. The characteristic of the radial stress and the circumferential stress both decrease gradually from the friction region to both sides of the disc. The maximum circumferential stress is 147 MPa at 38 s. The maximum radial stress is 120 MPa at 35 s. The maximum von-Mises stress and maximum friction temperature behaved inconsistently. Thermal stress increased rapidly at the previous stage of the braking phase and reached up 192 MPa at 38 s, which correspond to a relatively large temperature gradient.

Figure 16a–d present the von-Mises stress distributions of the disc at different stages of the braking process. The surface temperature rose faster than the brake disc body, resulting in different thermal expansions between the surface and body of the brake disc [29]. At 12 s of the initial braking stage (Figure 16a), the thermal stress is concentrated on the friction surface. The thermal stress is proportional to the temperature gradient. The maximum thermal stress reaches up to 192 MPa at the position of the bolt hole, the edge of the friction interface, and ribs (black arrow indicates), appeared at t = 38 s (Figure 16b). The maximum thermal stress of 192 MPa is still less than the allowable fracture strength 280 MPa of aluminum alloy. The thermal stress decreased gradually in the following steps of the braking process (Figure 16c). The low thermal stress of about 140 MPa appeared in the internal of the disc at 83 s (Figure 16d). In general, higher convection and conductivity of the Al alloy resulted in lower temperature and lower thermal stress distributions. Compared with the results from [27,28,29,30], fewer hot spots can be found in our study. It indicates that lower thermal stress and fewer thermal cracks are produced during the braking process, which relatively decreases the damage of the brake disc. 

### 5.4. Macrostructures and Microstructure of Al_2_O_3_-SiC_(3D)_/Al Alloy Composite

The macrograph image of Al_2_O_3_-SiC RPCs by pressureless sintering in air at 1450 °C for 1 h is shown in Figure 17a. The relative density of sintered Al_2_O_3_-SiC RPCs is about 95%. The pores are s array holes with pore diameters of 4 mm. It can be found from Figure 17b that molten Al alloy infiltrates the Al_2_O_3_-SiC RPCs completely. The darker phase is the Al alloy matrix, and the brighter phase is Al_2_O_3_-SiC RPCs. Both Al_2_O_3_-SiC RPCs reinforcement and Al alloy phase show the interpenetrating structure and are distributed homogeneously. Molten Al alloy infiltrating into the Al_2_O_3_-SiC struts has no significant effect on the interface (red dashed lines indicate). Accordingly, the layers are stacked over each other, where each layer is bonded with the adjacent layer deposition shown in Figure 17c. This is typical characteristic 3D-printed clay accumulation. The diagrammatic sketch of brake disc with layer is shown in Figure 17d.

The Al alloy matrix is characterized by the presence of α-Al dendrites, surrounded by the Si eutectic structure (little purple arrow indicates) shown in Figure 18. Coarse inter-metallic phases, containing Fe (little red arrow indicates), Cu (little green arrow indicates) formed during solidification and remained undissolved by T6 heat treatment, were also observed in the interdendritic region. The strength and hardness of Al alloy were improved because the supersaturated solid solution decomposes as time increases. Nevertheless, strengthening intermetallic precipitates induced by heat treatment, due to the nanometric size, cannot be detected by OM. Secondary dendrite arm spacing (SDAS) with the maximum size 40 μm is obtained by T6 heat treatment. 

### 5.5. Friction Experiment and Wear Mechanism

Figure 19 shows the friction experiment data of the braking ring against PM pins at IBV 97 m/s under 0.80 MPa. The predicted maximum friction temperature 517 °C is very close to the experiment measured result 457 °C. As the distance between the point for measuring temperature of thermocouple and the friction surface is 0.5 mm, the measured temperature is lower than the predicted maximum temperature. The torque curve exhibits the similar tendency of a “saddle” shape, namely a relatively smooth middle stage. The sharp peak appears at the end of the curve. The average friction coefficient is 0.31. The predicted braking time 83 s in FEM is longer than the measured result 75 s. The simulation method had the capability to provide accurate temperature prediction in the brake application. At the same time, reliable experiment data can be obtained by sub-scale testing, which can save lots of experimental funds and time [28].

Figure 20 shows the worn surface of the brake disc after testing at IBV 97 m/s under 0.80 MPa. The friction surfaces of the brake disc maintained a smooth level from Figure 20a. The brake ring had less wear mass loss since a mechanically mixed layer (MML) was formed on the friction surface, which effectively improves the wear resistance. The worn surface shows shallow unidirectional grooves conforming to the applied abrasive wear conditions. Some Al_2_O_3_-SiC struts are exposed in the fracture zones, indicating the interface of Al_2_O_3_-SiC struts and Al alloy might be a source of the delamination fracture [29]. The deepest part of the grooves is only 26 μm (Figure 20b) due to the supporting of high-strength and high-melting-point Al_2_O_3_-SiC struts. Al_2_O_3_-SiC struts rooted the soft matrix at the original position without serious distortion, and even the Al alloy matrix was torn by force of friction. The plastic deformation and softening of Al alloy were restricted as well. Some more micro-holes were found in the MML due to serious wear damage at IBV 97 m/s.

The cracks of the brake disc caused by the periodic thermal stress are along the radial directions. Thermal stress has a periodic effect on the brake disc during braking process, so the circumferential stress is the main factor that accounts for the initiation and propagation of a crack on the friction surface [30]. If thermal stresses repeat many times, brake disc failure is inescapable. A diagrammatic sketch of thermal tension stress on the circumferential and radial crack of brake disc is shown in Figure 21. In this study, we chose the Al_2_O_3_-SiC_(3D)_/Al alloy composite as wear-resisting surface layers for the brake disc. The radial cracks were observed on the friction surface after a long period of repeated thermal fatigue at higher IBV. The cracks are perpendicular to the siding direction. However, the grooves and cracks are terminated at the interface of Al_2_O_3_-SiC struts and Al alloy. It indicates that interface can prevent crack growth in the braking process. The experiment results indicated the disc can meet the requirement of the high-speed train with a speed 350 km/h. 

## 6. Conclusions

The FEM-CFD coupling calculation for Al alloy brake discs with Al_2_O_3_-SiC_(3D)_/Al alloy wear-resisting surface layer at IBV of 55−97 m/s was investigated and the conclusions can be drawn as follows:1)The sequential coupling method was adopted in our study to realize the coupled calculation method of frictional heat-stress-flow field.2)As *h_c_* calculated by CFD is a function of braking time, it can accurately predict the thermal stress concentration area in the brake disc.3)Fewer hot spots can be found for the brake disc with Al_2_O_3_-SiC_(3D)_/Al alloy wear-resisting surface layers. Lower thermal stress and fewer thermal cracks were produced on the friction surface during the braking process, which relatively decreases the damage of the brake disc.4)The reliable experiment data can be obtained by sub-scale testing.

## Figures and Tables

**Figure 1 materials-12-03155-f001:**
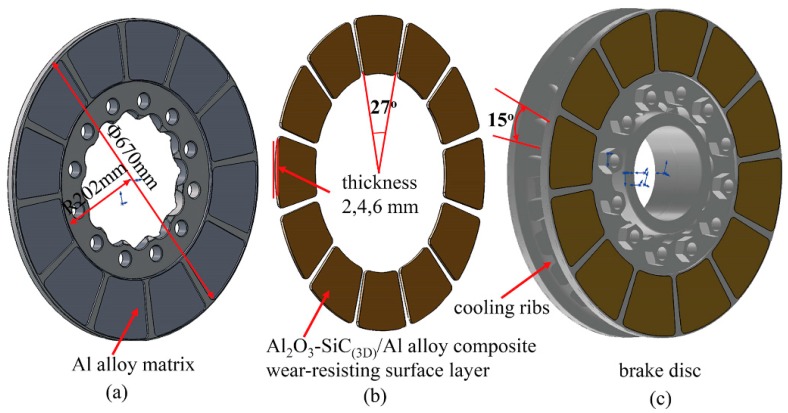
3D brake disc model for calculation: (**a**) Al alloy matrix of brake disc; (**b**) Al_2_O_3_-SiC_(3D)_/Al alloy composite wear-resisting surface layer, thickness 2, 4, and 6 mm; (**c**) Al alloy brake disc.

**Figure 2 materials-12-03155-f002:**
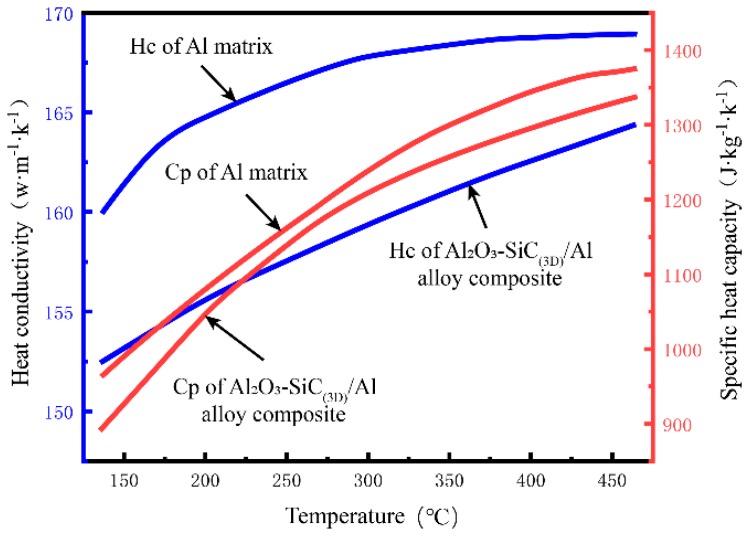
Heat conductivity and specific heat capacity of brake disc materials.

**Figure 3 materials-12-03155-f003:**
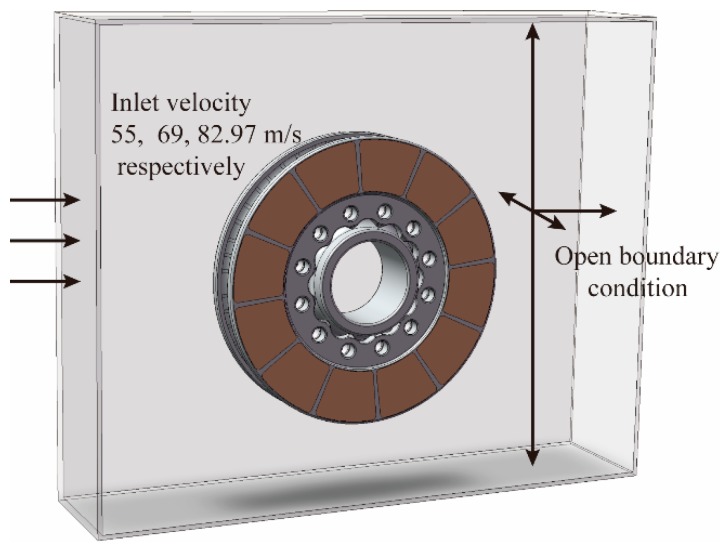
Computational fluid dynamic (CFD) model for air convection calculation.

**Figure 4 materials-12-03155-f004:**
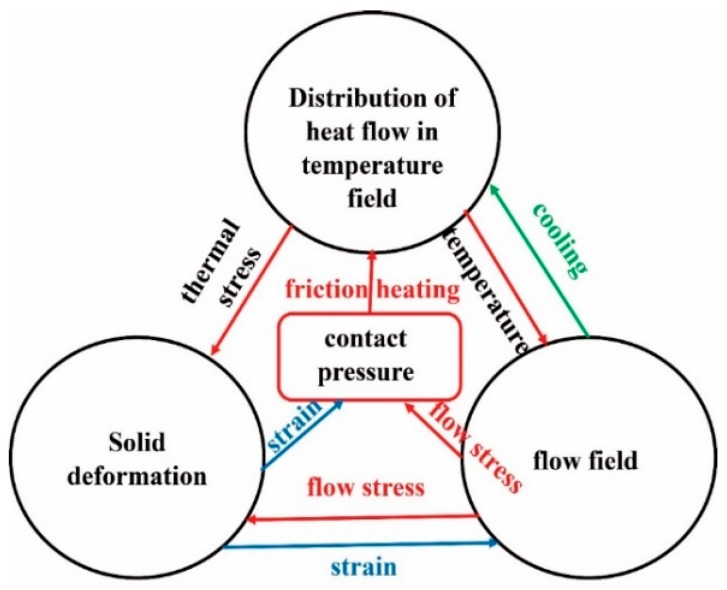
Multi physical field coupling diagram of the disc brake.

**Figure 5 materials-12-03155-f005:**
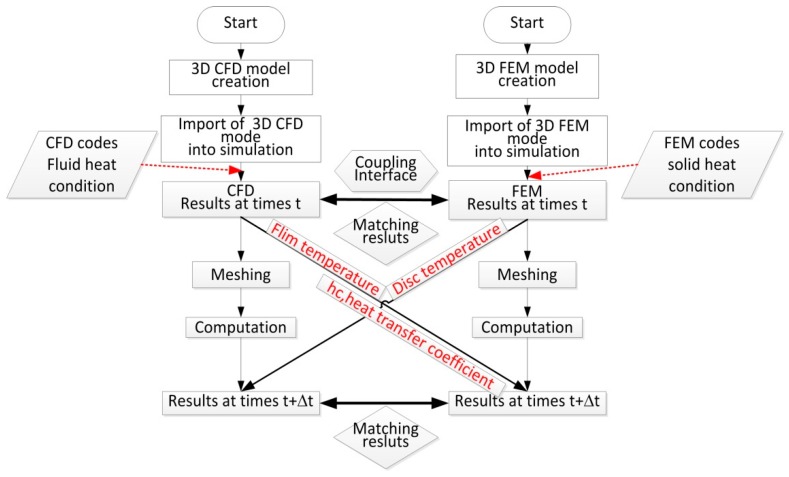
Finite element method (FEM)-CFD coupling calculation flow chart of disc brake.

**Figure 6 materials-12-03155-f006:**
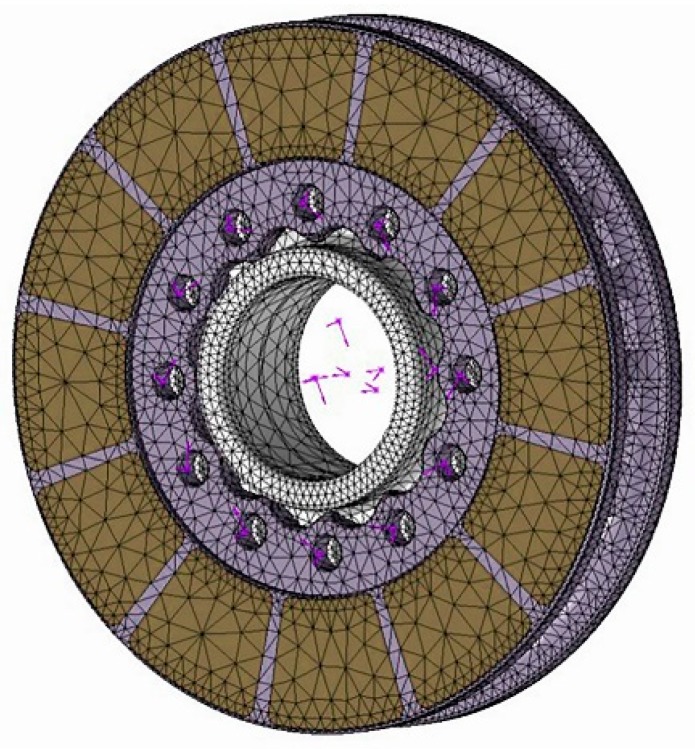
Meshing of the 3D thermal-stress calculation model.

**Figure 7 materials-12-03155-f007:**
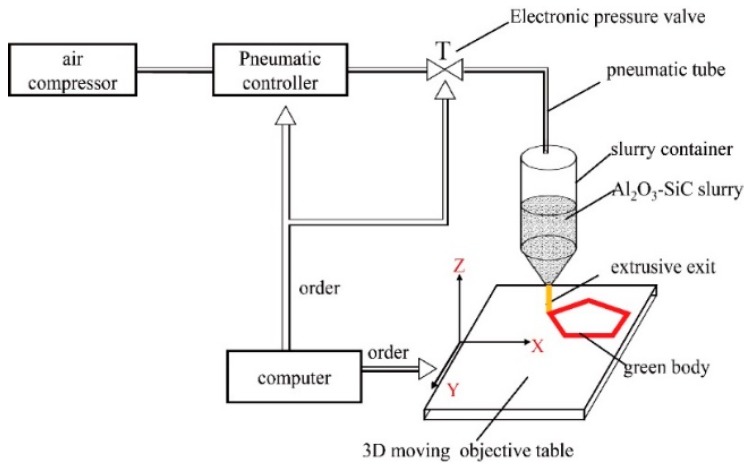
Schematic diagram of the 3D extrusion printing machine for fabricating Al_2_O_3_-SiC_(3D)_ green bodies.

**Figure 8 materials-12-03155-f008:**
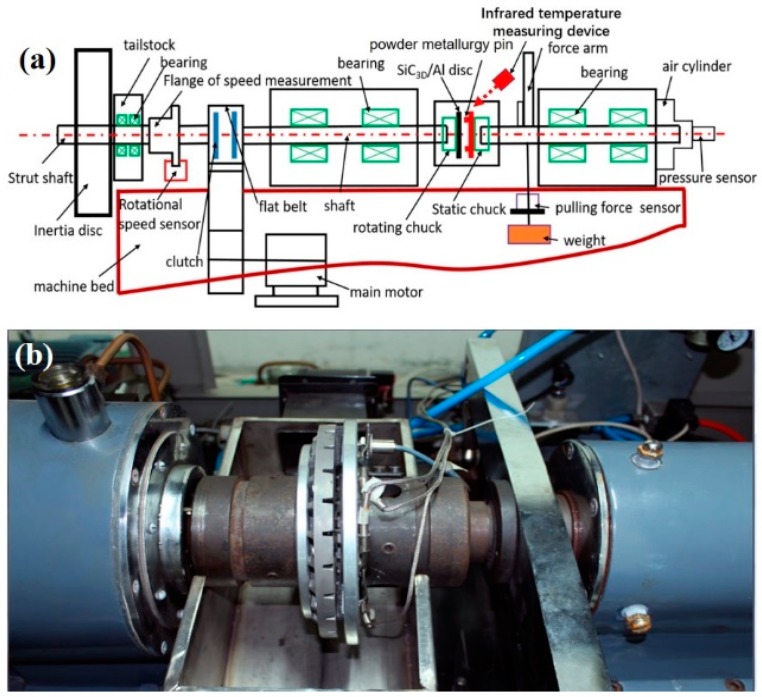
MM3000 sub-scale testing machine: (**a**) schematic diagram; (**b**) photo of experimental equipment.

**Figure 9 materials-12-03155-f009:**
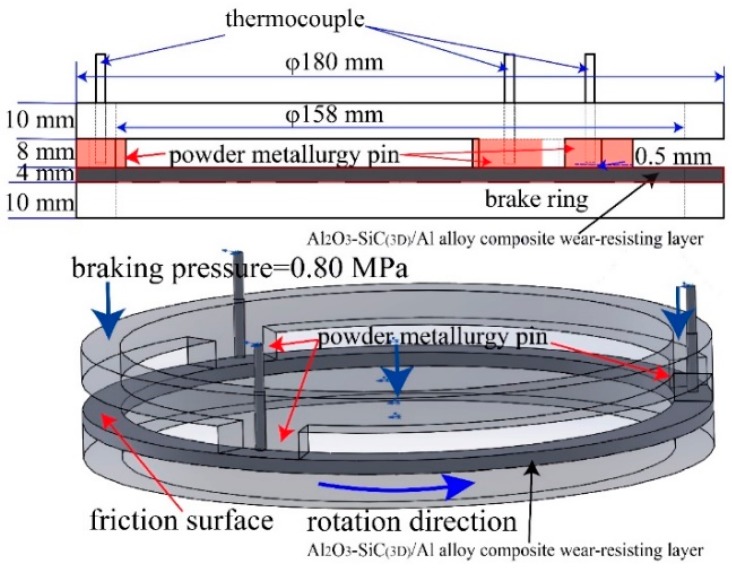
Schematic diagram of brake ring and powder metallurgy pins used for the sub-scaled test.

**Figure 10 materials-12-03155-f010:**
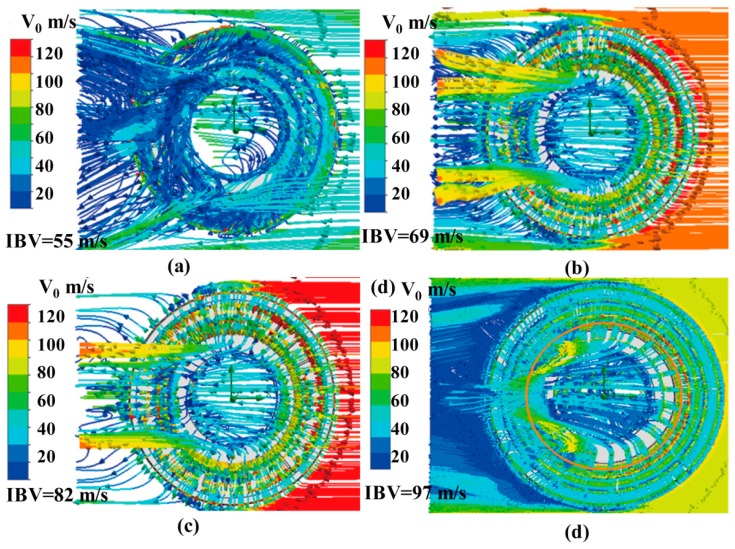
3D streamlines of the air around brake disc at t = 5 s after start of braking: (**a**) initial braking velocity (IBV) = 55 m/s; (**b**) IBV = 69 m/s; (**c**) IBV = 82 m/s; (**d**) IBV = 97 m/s.

**Figure 11 materials-12-03155-f011:**
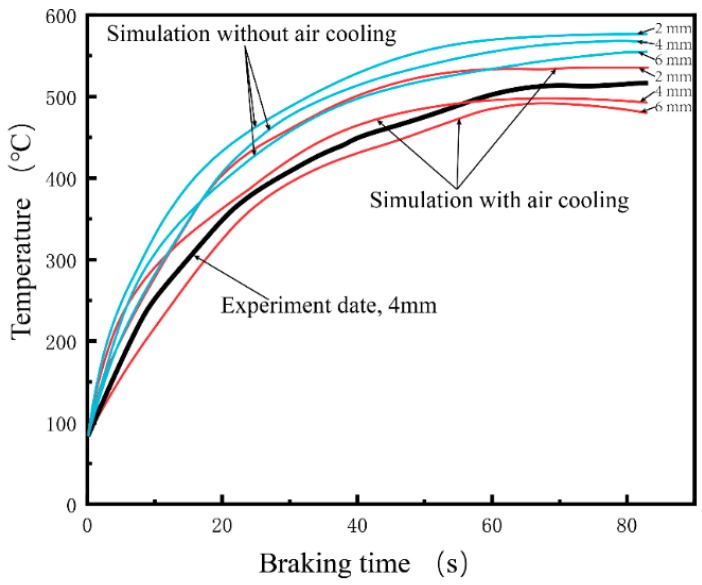
Variation of the maximum friction temperature on the wear-resisting layers of the brake discs with different thickness of wear-resisting layer versus braking times (IBV = 97 m/s).

**Figure 12 materials-12-03155-f012:**
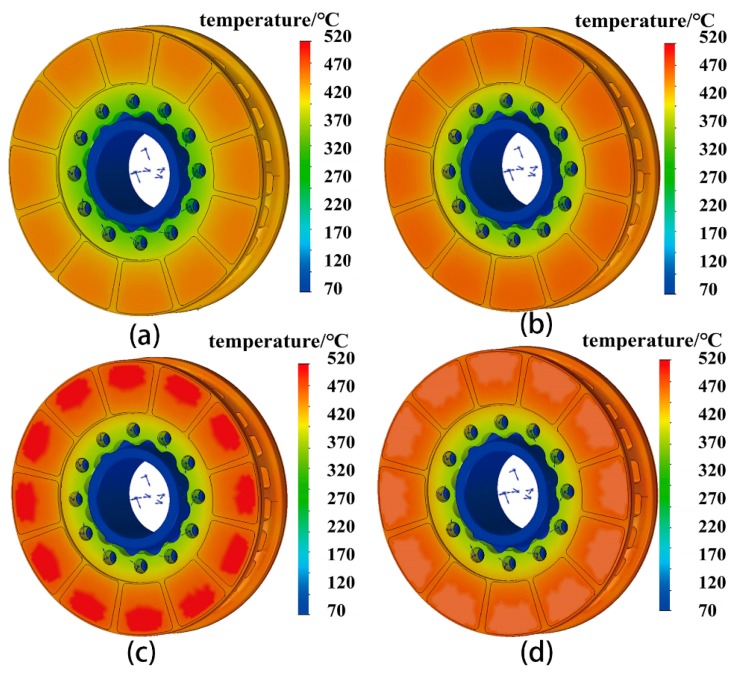
Temperature distribution of the brake disc at different stages of braking operation (IBV = 97 m/s, thickness of wear-resisting layer = 4 mm): (**a**) 12 s; (**b**) 30 s; (**c**) 63 s; (**d**) 83 s.

**Figure 13 materials-12-03155-f013:**
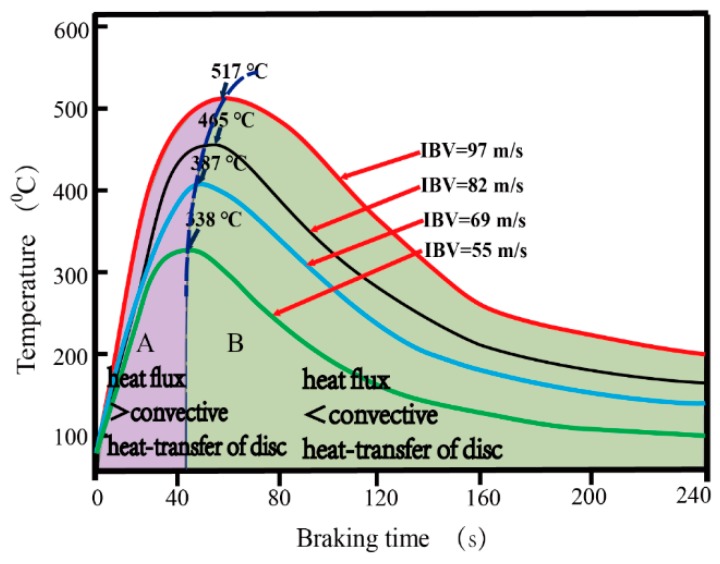
Relationship between transient temperature and IBV of 55~97 m/s under 22 kN.

**Figure 14 materials-12-03155-f014:**
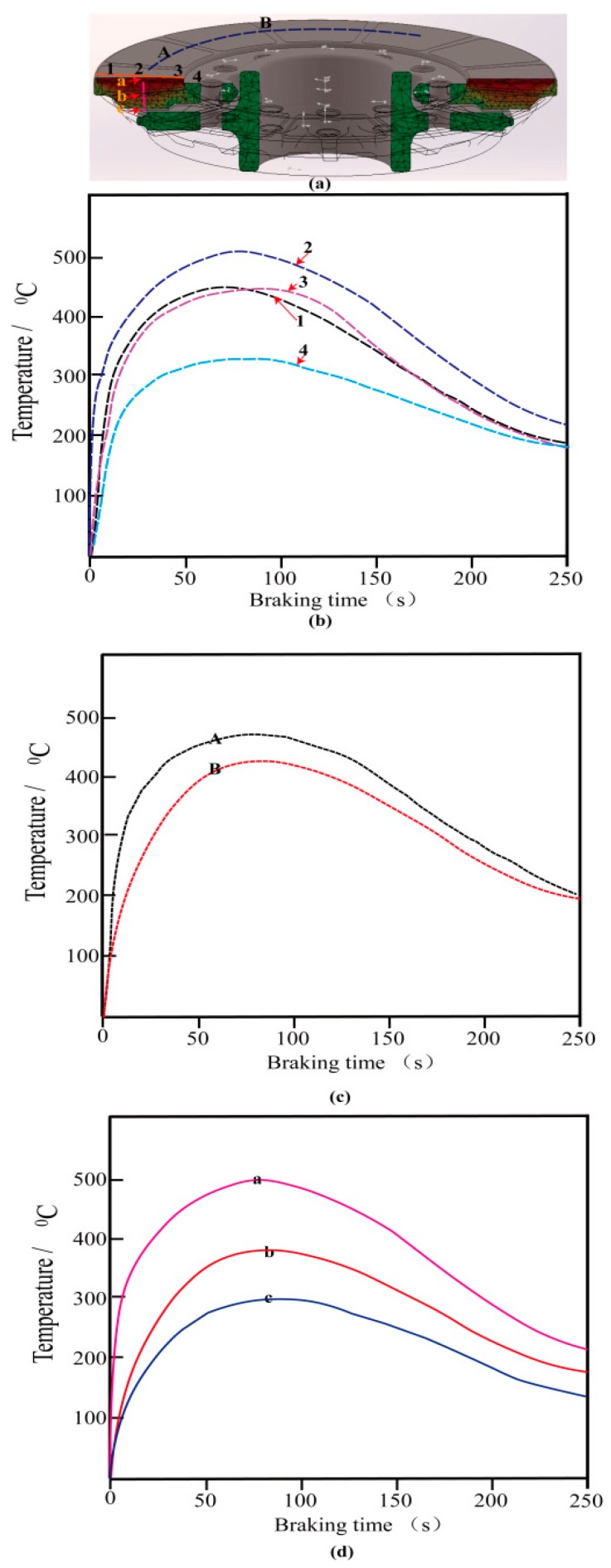
Temperature in the circumference and radial direction versus time at different radii (IBV = 97 m/s, thickness of wear-resisting layer = 4 mm): (**a**) brake disc; (**b**) 1–4 points; (**c**) A,B points; (**d**) a,b,c points.

**Figure 15 materials-12-03155-f015:**
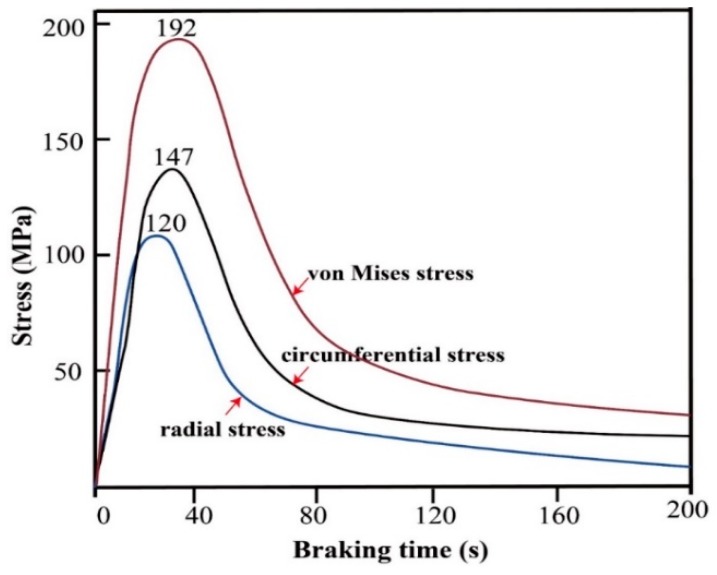
Stress in the circumference and radial direction versus time at different radii (IBV = 97 m/s, thickness of wear-resisting layer = 4 mm).

**Figure 16 materials-12-03155-f016:**
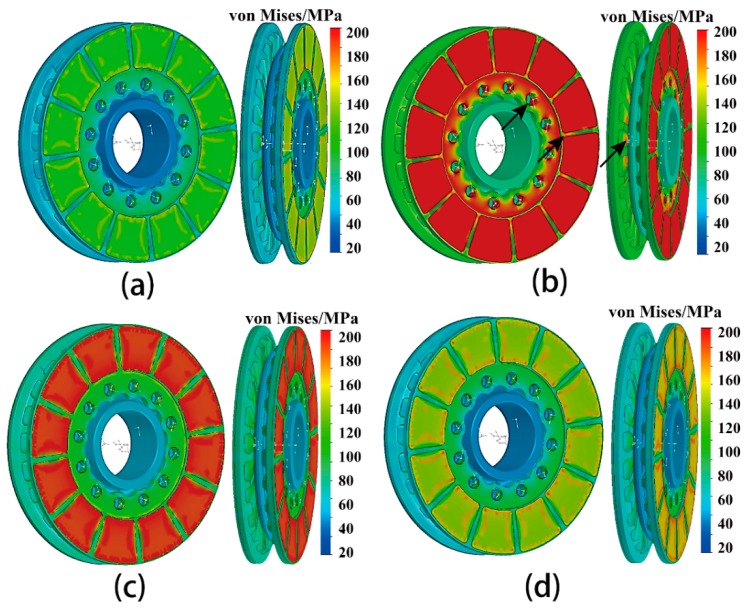
Stress distribution of the brake disc in the brake disc at different stages of braking operation (IBV = 97 m/s, thickness of wear-resisting layer = 4 mm): (**a**)12 s; (**b**) 38 s; (**c**) 46 s; (**d**) 83 s.

**Figure 17 materials-12-03155-f017:**
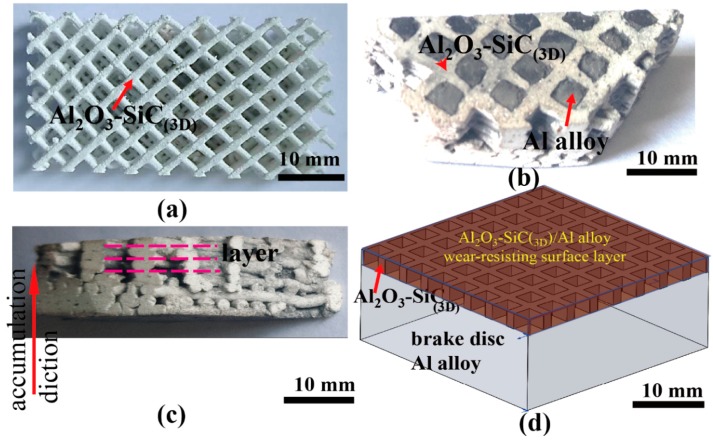
Macrostructures of Al_2_O_3_-SiC reticulated porosity ceramics (RPCs) and Al_2_O_3_-SiC_(3D)_/Al alloy: (**a**) sintered Al_2_O_3_-SiC RPCs; (**b**) fracture morphology of Al_2_O_3_-SiC_(3D)_/Al alloy; (**c**) side view of Al_2_O_3_-SiC_(3D)_ RPCs; (**d**) diagrammatic sketch of brake disc with layer.

**Figure 18 materials-12-03155-f018:**
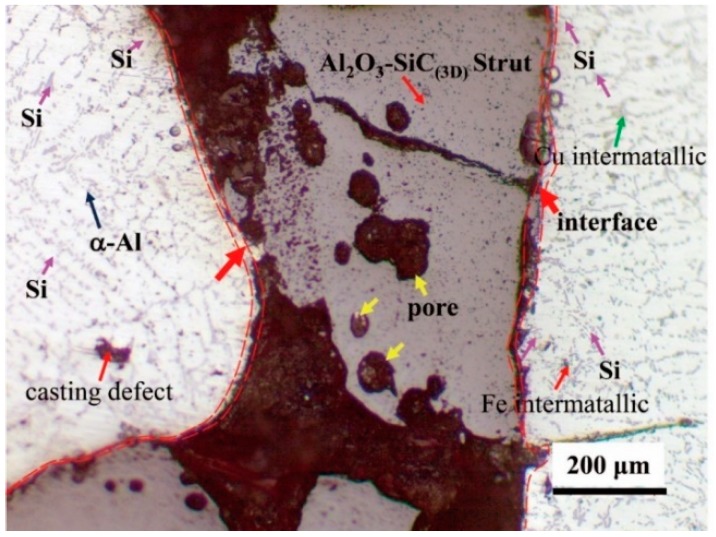
Microstructure of Al_2_O_3_-SiC_(3D)_/Al alloy composite.

**Figure 19 materials-12-03155-f019:**
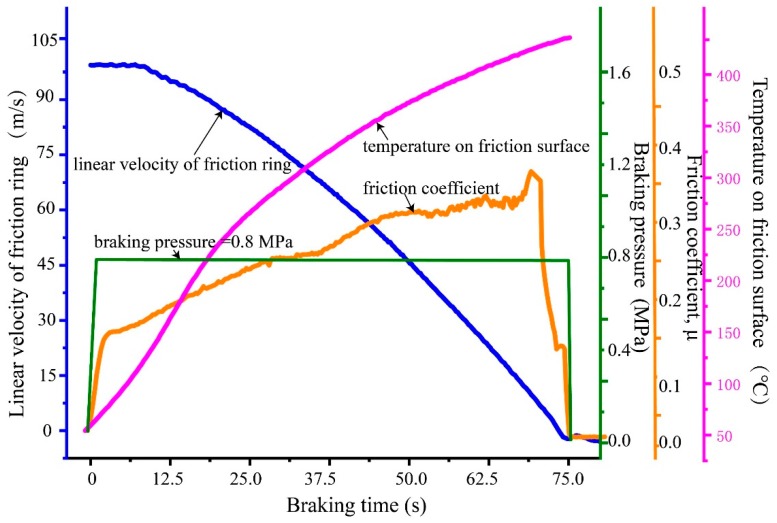
Friction experiment data of the braking ring against powder metallurgy (PM) pins at IBV 97 m/s and 0.80 MPa.

**Figure 20 materials-12-03155-f020:**
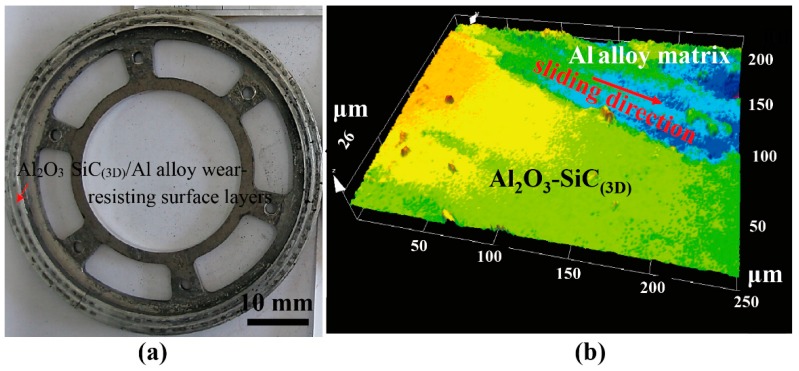
Worn surface of the brake ring after testing at IBV = 97 m/s and pressure 0.80 MPa: (**a**) macrostructures of the brake ring; (**b**) worn surface tested by laser scanning confocal microscope.

**Figure 21 materials-12-03155-f021:**
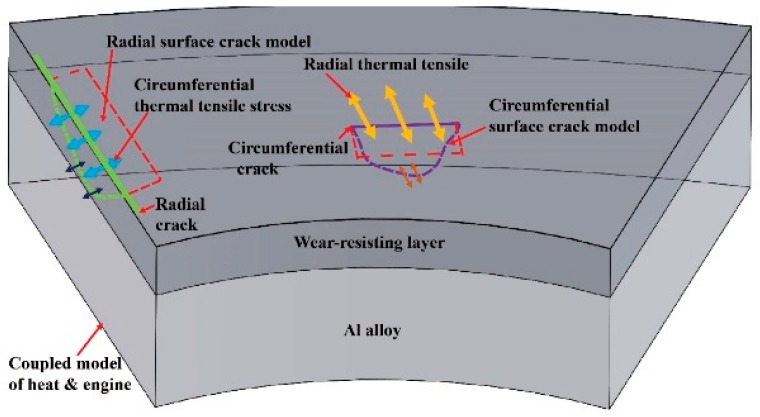
Diagrammatic sketch of thermal tension stress on circumferential and radial cracks.

**Table 1 materials-12-03155-t001:** Properties of brake disc material.

Materials	Elastic Modulus E/(GPa)	Poisson ν	Density ρ/(kg·m^−3^)
Al_2_O_3_-SiC_(3D)_/Al	248	0.18	3700
Al alloy matrix	69	0.27	2700

**Table 2 materials-12-03155-t002:** Basic data of the brake disc.

Parameters	Values
Shaft mass of single disc, M/kg	4.4 × 10^3^
IBV, V_0_/(m·s^−1^)	55, 69, 82, 97
Deceleration, a/(m·s^−2^)	1.17
Friction coefficient of disc/pad, μ	0.32
Contact friction area (both sides), A (mm^2^)	4.48 × 10^5^
Mass of the single disc, m/(kg)	22
Force applied braking pressure, F_b_/(kN)	22
Radius of brake discs, r_d_ (mm)	335
Radius of wheel, r_w_ (mm)	455

**Table 3 materials-12-03155-t003:** Experimental parameters for sub-scale testing.

Brake Disc Initial (IBV) km/hour	Braking Inertia (kg·m^2^)	Brake Disc Max Radius Linear Speed (m·s^−1^)	MM3000 Machine Angular Velocity (rad/s)	MM3000 Machine Angular Velocity (r/min)	Energy Absorbed by the Unit Area (J·mm^−2^)
200	0.80	55	656	6267	13.36
250	0.80	69	816	7796	21.04
300	0.80	82	970	9267	29.72
350	0.80	97	1147	10958	41.58

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
