# Peer review of "Thermo-Mechanical Coupling Analyses for Al Alloy Brake Discs with Al2O3-SiC(3D)/Al Alloy Composite Wear-Resisting Surface Layer for High-Speed Trains"

_materials, 2019, doi:10.3390/ma12193155_

Round 1

Reviewer 1 Report

Dear Authors,

1. There are numerous places in the text with English grammatical errors. The authors should be full checking for grammar and mistakes to meet the quality of Journal. 

2. Add the list of symbols. 

3. Most equations in the paper, the authors don't mention to any references. Add citation to the equations which took from references.

4. Add a new flowchart to explain the details of the selected approach.

5. How you obtain the results of figure 2, explain with details.

6. There are many research papers study the same problem which investigated in the present paper. What is exactly the new point of this work?

The authors should focus to clarify this issue in the paper.

7. The meaning of the conclusions is unclear, and there are grammatical errors. The authors should think over the real significance of their results and try to rewrite this section to improve understanding of the conclusions.

Altogether, the paper needs modification to be suitable for the standards required for publication; therefore I recommend that it required to major revision.

Author Response

Dear reviewers:

Thank you for your letter and the reviewers’ comments on our manuscript ID: materials-527666 entitled " Thermo-mechanical coupling analyses for Al alloy brake disc with Al2O3-SiC(3D)/Al alloy composite wear-resisting surface layer for high-speed train". Those comments are very helpful for revising and improving our paper, as well as the important guiding significance to other research. We have studied the comments carefully and made corrections which we hope meet with approval. The main corrections are in the manuscript and the responds to the reviewers’ comments are as follows (the replies are highlighted in red).

Most sincerely,

YU L

August 15 ,

Reviewer 2 Report

The proposed paper deals with thermo-mechanical coupling analysis of a disc-pad system, with the disc made of Al2O3-SiC(3d)/Al composite material

first simulations are developed based on a proprietary software. Then experiments are presented which tend to agreed with the simulations. Analysis of the adequation between simulations and experiments is developed.

my comments :

for the simulations the energetic approach is used to bypass contact problem in FEM and define thermal boundaries

it is not clear for me how has been linked the purely energetic approach for the simulation with the experimental setup to impose the same heat flux at the contact surface. This is the key problem to compare previous simulations with experimental measures, and need its own section in such article ! maybe before section 4.3

Can you present the MM300 machine ?

on the other hand, the FEM simulations are not based on a composite structure such as the one presented in figure 15, can you explain why ?

figure 12 : statistical results are expected when dealing with experiments to be compared with FEM, else it is supposed to be 'good luck' ! please provide mean and standard deviations of measures, as well as sensors properties. Maybe spliting fig.12 into 4 pictures migth help understanding

Author Response

(The authors gave the same response as above.)

Round 2

Reviewer 1 Report

The authors made all required corrections. Best regards, Reviewer